# Noninvasive Monitoring of Subsurface Soil Conditions to Evaluate the Efficacy of Mole Drain in Heavy Clay Soils



Akram Aziz [1], Ronny Berndtsson [2],*, Tamer Attia [1], Yasser Hamed [3] and Tarek Selim [3],*

1 Geology Department, Faculty of Science, Port Said University, Port Said 42522, Egypt
2 Centre for Advanced Middle Eastern Studies & Division of Water Resources Engineering, Lund University, SE-22100 Lund, Sweden
3 Civil Engineering Department, Faculty of Engineering, Port Said University, Port Said 42523, Egypt
* Correspondence: ronny.berndtsson@tvrl.lth.se (R.B.); eng_tarek_selim@yahoo.com (T.S.)

**Abstract:** Soil degradation and low productivity are among the major agricultural problems facing farmers of the newly reclaimed agricultural area in the Nile Delta region, Egypt. High content of clay and silt characterizes the soil texture of all farms in the area, while farmers still rely on the traditional mole drainage (MD) system to reduce the salinity of the farm soil. We present a comparison of innovative geo-resistivity methods to evaluate mole drains and the salinity affected clay soils. Geoelectrical surveys were conducted on three newly reclaimed farms to image the subsurface soil drainage conditions and to evaluate the efficiency of using the traditional MD systems in these heavy clay environments. The surveys included measuring the natural spontaneous potential (SP), apparent resistivity gradient (RG), and electrical resistivity tomography (ERT). Integrating the results of the three methods reduced the ambiguous interpretation of the inverted ERT models and allowed us to determine the subsurface soil structure. The inverted ERT models were suitable for locating the buried MDs and delineating the upper surface of the undisturbed clay beds. The proximity of these layers to the topsoil reduces the role played by MDs in draining the soil in the first farm and prevents the growth of deep-rooted plants in the second farm. Time-lapse ERT measurements on the third farm revealed a defect in its drainage network where the slope of the clay beds opposes the main direction of the MDs. That has completely obstructed the drainage system of the farm and caused waterlogging. The presented geo-resistivity methods show that integrated models can be used to improve the assessment of in situ sub-surface drainage in clay-rich soils.

**Keywords:** water logging; resistivity gradient; ERT; electrical conductivity; soil drainage

## 1. Introduction

Most countries in the Middle East include large arid and semi-arid climate regions [1]. They all face the challenge of securing enough food supplies in unfavorable dry climate conditions with water scarcity. They need not only to conserve their existing resources (e.g., water, cultivated areas, etc.) but also to maximize the utilization of their resources. In Egypt, the government has launched a national project for land reclamation to meet the increasing food demands of its 110 million population. The area south of Port Said city (Figure 1a,b) is one of the main sectors in that project, as its production serves the districts of the Suez Canal and northern Sinai regions [2,3]. This area was once a fertile region in ancient Egypt [4–6]. Currently, the area is characterized by low precipitation—around 150 mm annually—and high evapotranspiration rates. The temperature ranges from 31° to 36 °C during July/August, and from 8° to 19 °C during December/January [7]. Local farmers mainly rely on the use of the traditional mole drain (MD) method to increase the drainage of the clay-rich soils. Mole drains are sets of unlined soil channels dug by a mole plow. The plow consists of a cylindrical foot attached to a narrow shank followed by a cylindrical expander. The shank creates a narrow slot extending from the soil surface down

to the mole channel. Each channel is 10 cm in diameter and 50 to 70 cm deep below the soil surface. Heavy soils with low hydraulic conductivity need a closely spaced drainage system (2–4 m). The MD technique is used in combination with open surface drains at the boundaries of the reclaimed farms to facilitate the soil internal drainage, minimize deep percolation, and control groundwater table level, thereby increasing crop production.

Recently, many of the newly reclaimed farms within this area are facing serious agricultural problems that have led to soil degradation [8–11]. Previous studies [12,13] have shown that about 35% of the Egyptian irrigated lands suffer from salinization problems due to improper agricultural practices and inefficient drainage systems. In view of this, there is an urgent need to develop simple and non-invasive techniques for the evaluation of existing drainage systems and the state of salinity in the heavy clay soils. Thus, the objective of this study was to compare three innovative geo-resistivity methods for the clay-rich and salinity-affected farm soils. Since salinity may affect the readings from the different resistivity methods, it important to establish which methods give the most reliable results for varying salt content of the soil and how readings from different methods can be combined to improve results.

## 2. Methods & Materials

### 2.1. Site Description and Location

The geological map in Figure 1a shows that Holocene silty clay sediments cover the entire region of the Nile Delta. Moreover, as shown in Figure 1b, the location of the area under investigation in this research is bounded from the east by the Suez Canal and in the west by El-Manzala Lake, where sabkha deposits and salt crusts dominate the region. The experimental area is located between latitude 31°06′30″ and 31°06′50″ N, and longitude 32°17′15″ E and 32°17′50″ E (Figure 1c). The high content of clay in soils located within the Nile Delta region complicates water seepage and the provision of efficient drainage and sustainable soil productivity [14].

Three farms A–C located within the reclamation area were chosen to conduct the ground geophysical surveys aiming to decipher the subsurface conditions that may affect their productivity and to monitor the lateral and vertical variations occurring during different seasons. The vegetation cover on the first farm (Farm A) is heterogeneous, with plants dying after a short period of growth in certain areas (Figure 2A). No agronomic problems can be observed on the second farm (Farm B) except for the inability of deep-rooted vegetables to grow (Figure 2B). On the third farm (Farm C), all attempts to cultivate have failed because the soil constantly suffers from waterlogging problems (Figure 2C).

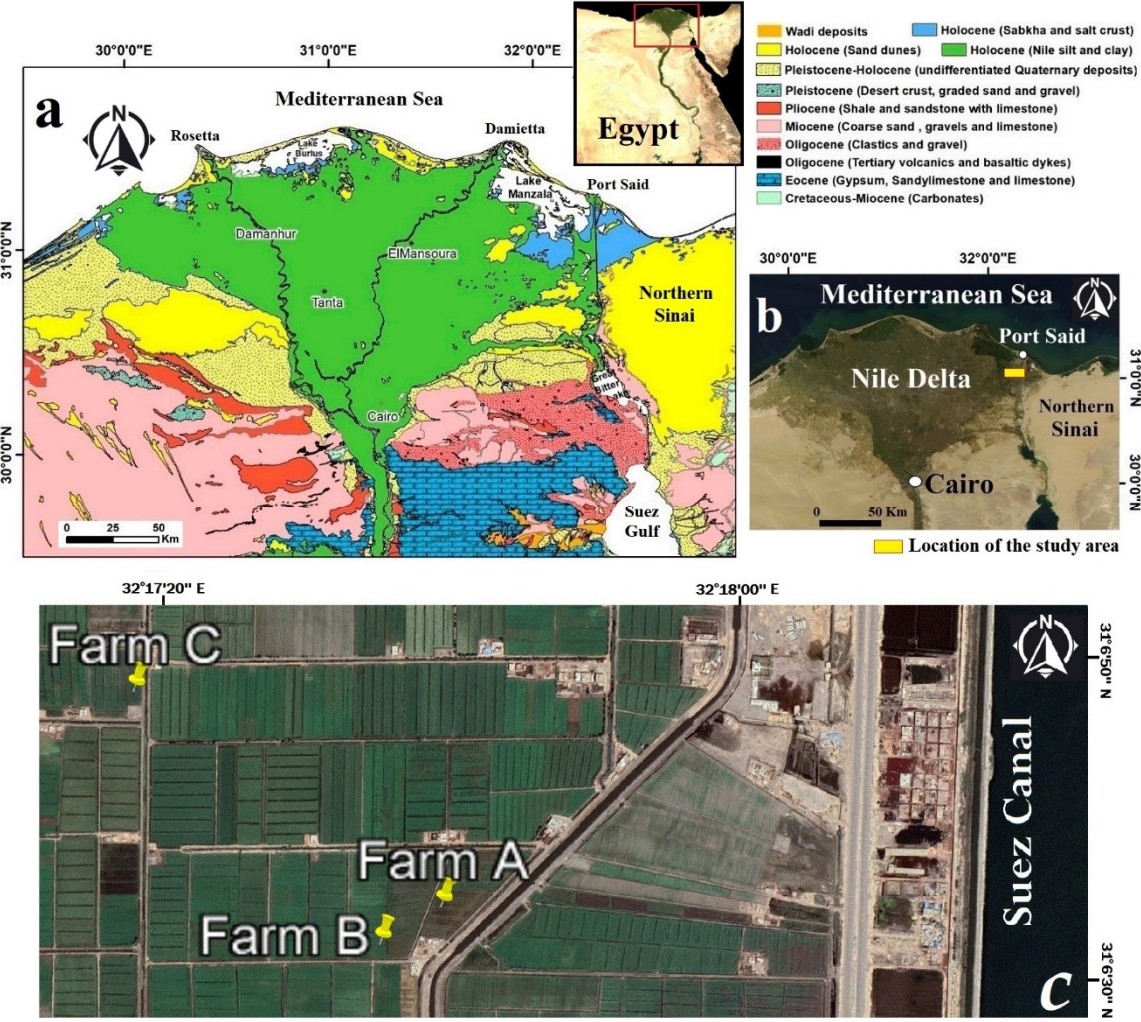

**Figure 1.** (**a**) Geological map of the Nile Delta (modified after Conoco [15] and Hassan et al. [16]), (**b**) location map of the recently reclaimed area denoted by the white rectangle, and (**c**) location of the three farms.

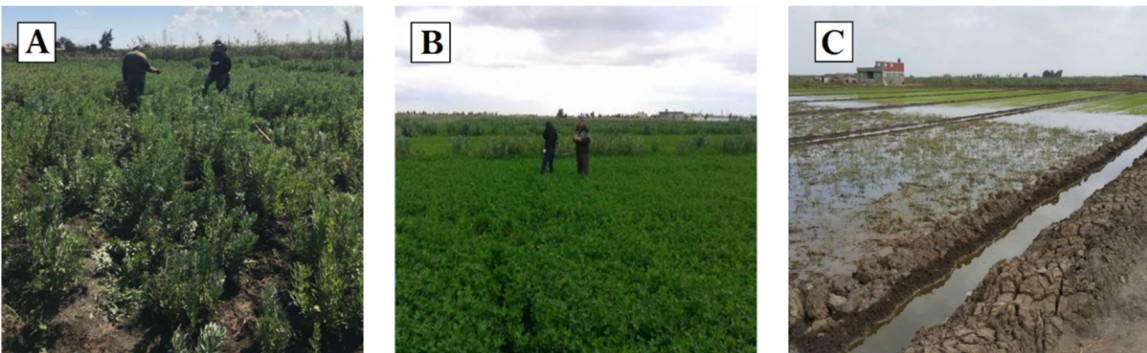

**Figure 2.** Agricultural conditions of the three investigated farms: (**A**) Heterogeneous and partially dead vegetation cover, (**B**) no deep-rooted plants can successfully grow, and (**C**) barren saline soil and frequent waterlogging.

### 2.2. Methodology

Geoelectrical methods have a wide range of applications in agriculture [17–20] because of their ability to monitor lateral and vertical variations in soil conductivity as a

function of the electrical resistivity [21–23]. They have been efficiently used for imaging clay-pan [24,25], peatland stratigraphy [26], root zones [27–31], and for monitoring contamination [32,33]. They have also been applied in various post-reclamation problems [34], soil characterization [35,36], the monitoring of clay behavior during seasonal water content variations [37,38], and various other near-surface investigations [39]. In this study, three geoelectrical methods, namely electrical resistivity tomography (ERT), spontaneous potential (SP), and apparent resistivity gradient (RG), were employed.

2.2.1. Electrical Resistivity Tomography (ERT)

This method relies on inverting the apparent resistivity data, measured along a profile, to create a 2D model that shows vertical and horizontal resistivity changes underneath the surveyed soil surface [40]. Because of the high background noise levels in cultivated soils, the Wenner-alpha array was chosen to implement the survey. The array is less sensitive to noise contamination because it has the highest signal strength compared to other arrays [41]. It is very sensitive to vertical changes in apparent resistivity. Therefore, it can resolve subsurface horizontal structures even at noisy sites [40]. Figure 3 shows the design of the ERT survey used in the current study. It consisted of 26 electrodes arranged along a profile and spaced by a unit distance ($a$). The apparent resistivity ($\rho_a$) of each station was calculated according to:

$$\rho_{a(\Omega.m)} = 2\pi a \frac{\Delta V}{I} \tag{1}$$

where $I$ is the intensity of direct current introduced into the ground via electrodes A and B and $\Delta V$ is the potential difference measured between the inner electrodes M and N (Figure 3). The survey starts by measuring apparent resistivity along the profile, using the Wenner-alpha array with electrode spacing of '$1a$'. The profile is re-surveyed several times. Each time the spacing between electrodes is increased by n factor to map different soil depths [42]. As the spacing between electrodes increases, the number of measurements will decrease and consequently the width of the apparent resistivity pseudo-section (Figure 3). The whole set of apparent resistivity measurements is checked to remove abnormal (noise) data. Noise is removed in the processing steps before final model calculations. Then the data can be inverted using RES2DINV software [43,44] to construct ERT models.

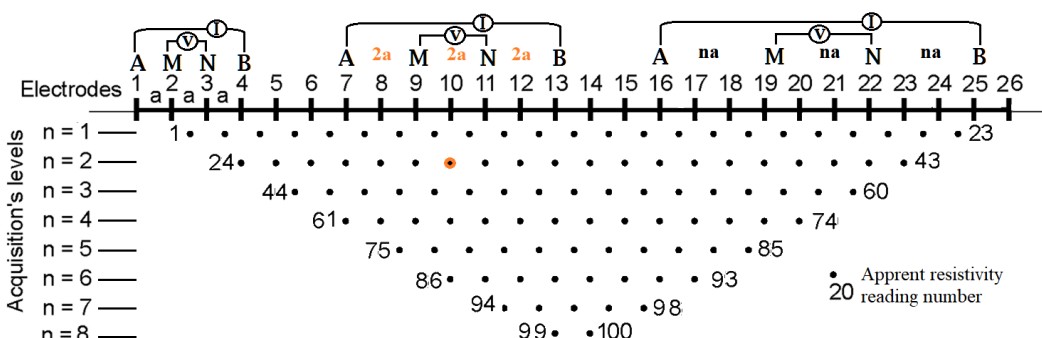

**Figure 3.** Schematic of electrode arrangement and the sequence of measurements used to build apparent resistivity pseudo-sections.

The inversion process aims to determine the true resistivity of the subsurface layers from the apparent resistivity pseudo-section. Thus, the process begins with constructing apparent resistivity pseudo-sections from a series of readings measured at the ground surface by using finite-difference or finite-element methods to divide the section into many rectangular blocks with different resistivity values [40,45]. Then the Jacobian matrix of partial derivatives is calculated using Gauss–Newton or quasi-Newton methods. Finally, the process ends with solving the least-square equation of the Gauss–Newton and quasi-

Newton methods using a regularized least-square optimization method [43,44]. Two algorithms are used to constrain the regularized least-square optimization. The first is the blocky constrained least-square method (L1-norm); the second is the smooth constrained least-square method (L2-norm) [46].

Some ERT profiles were repeated at different times during the field investigations to monitor changes that occurred between different seasons. Before inverting these time-lapse measurements, it was necessary to remove the effect of temperature changes and express all apparent resistivity values at a standardized temperature of 25 °C as soil conductivity increases by 1.9% [47] to 2.02% [48] per one-degree Celsius increase in temperature [49,50]. The following equation [47,51] was used to remove the effect of temperature changes from apparent resistivity data:

$$\rho_{25\,°C} = \rho_{a_T} / \left[04470 + 1.4034 e^{(T/26.815)}\right] \tag{2}$$

where $\rho_{a_T}$ is the apparent resistivity measured at a temperature T in °C, and $\rho_{25\,°C}$ is the standardized apparent resistivity referenced to 25 °C.

Inverting these time-lapse datasets should be carried out using a joint inversion technique where the model obtained from the initial dataset is used to constrain the inversion of the later time datasets [45].

### 2.2.2. Spontaneous Potential (SP)

This is a passive geophysical method used for mapping galvanic currents that are naturally developed in soils. These currents originate either due to the difference in concentrations between soil fluids (diffusion or electrochemical potential), movement of fluids in porous soils (streaming or electrokinetic potential), electronic conduction occurring in sulfide ores (mineralization potential), or due to the presence of clay that adsorbs ions on its surface (shale potential) [32,52–55]. The low cost of equipment required to conduct SP surveys in the field is a major advantage. Only a portable high impedance millivolt meter and an insulated wire reel are needed to measure potential differences directly between two non-polarizable electrodes [39,51,56]. Non-polarizable electrodes are copper rods immersed in porous pots filled with a saturated copper sulfate solution (Figure 4).

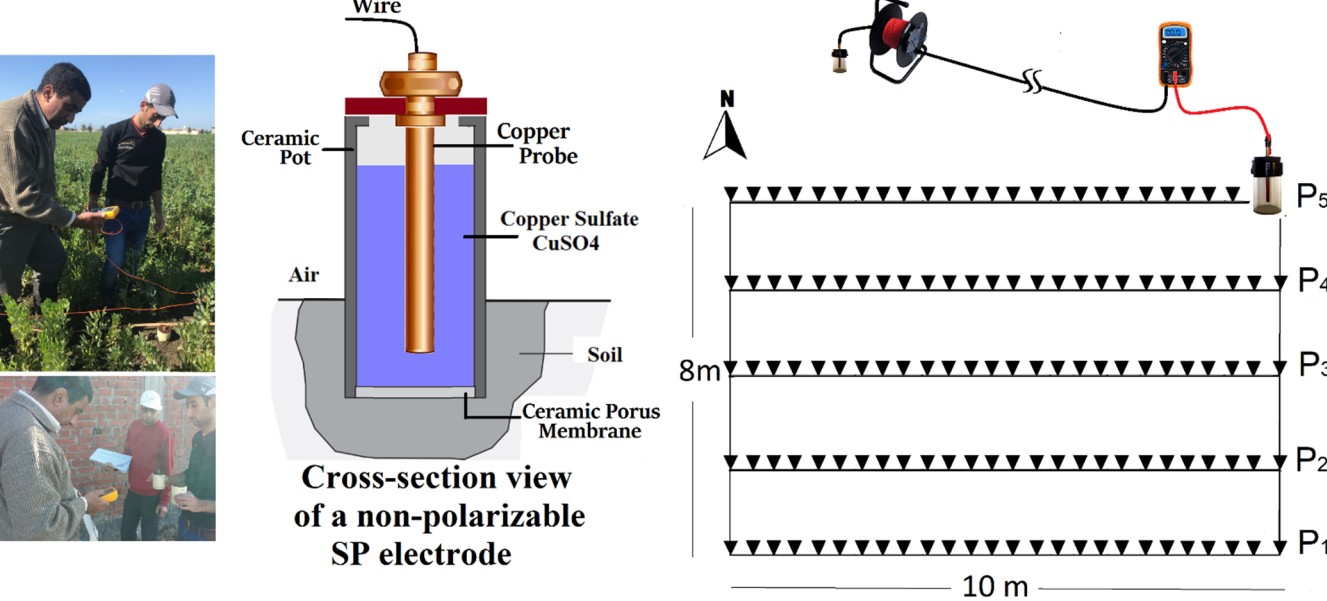

**Figure 4.** Placing the roving non-polarizable electrode in small well-watered holes to ensure efficient contact with the soil and layout of the grid used to conduct the SP survey using a fixed-based configuration.

One of these electrodes will be set as the remote reference station for all readings, while the other electrode will move along parallel profiles. For better results, the SP profiles should lay perpendicular to the direction of the subsurface flow [57]. After the roving electrode is implanted at each station, several minutes are allowed to pass during which the electrode stabilizes. Then the potential difference is measured relative to the reference electrode. During the work, contact resistance between the two electrodes should be tested regularly; a base station must be reoccupied with every constant interval of time for noise reduction. Linear corrections are applied to compensate for the effects of leakage or increased solution temperature inside the pots [58].

### 2.2.3. Apparent Resistivity Gradient

The apparent resistivity gradient (RG) method uses two metal electrodes, A and B, to introduce a direct current (DC) of intensity (I) into the ground. The current electrodes are kept at fixed positions while the voltage difference (v) is measured between another two rover electrodes M and N (Figure 5). After each measurement, only the voltage electrodes will leap by a distance equal to MN to measure voltage difference at the next stations (Figure 5). The acquisition procedures will be repeated along a set of profiles laying parallel to AB [59]. The apparent resistivity ($\rho_a$) in ohm meter ($\Omega$m) at each station is calculated according to [60]:

$$\rho_{a_{(\Omega.m)}} = \frac{2\pi V}{I}\left[\frac{1}{AM} - \frac{1}{BM} - \frac{1}{AN} + \frac{1}{BN}\right]^{-1} \tag{3}$$

where AM, BM, AN, and BN are the distances between electrodes at each station. All measurements are conducted within a rectangular area centered on the midpoint between A and B (Figure 5). The width and the length of the surveyed rectangular area should not exceed 0.3 and 0.5 of the distance between A and B, respectively, where the current electrodes produce a quasi-homogeneous field [61].

It is worth mentioning here that all of the ERT and RG surveys were performed using the IRIS Syscal-R2 resistivity instrument. The injection current duration was set to the maximum of 2000 ms and the stack (min/max) numbers were set to (3/10) to ensure the best quality of measurements in the highly conductive soil.

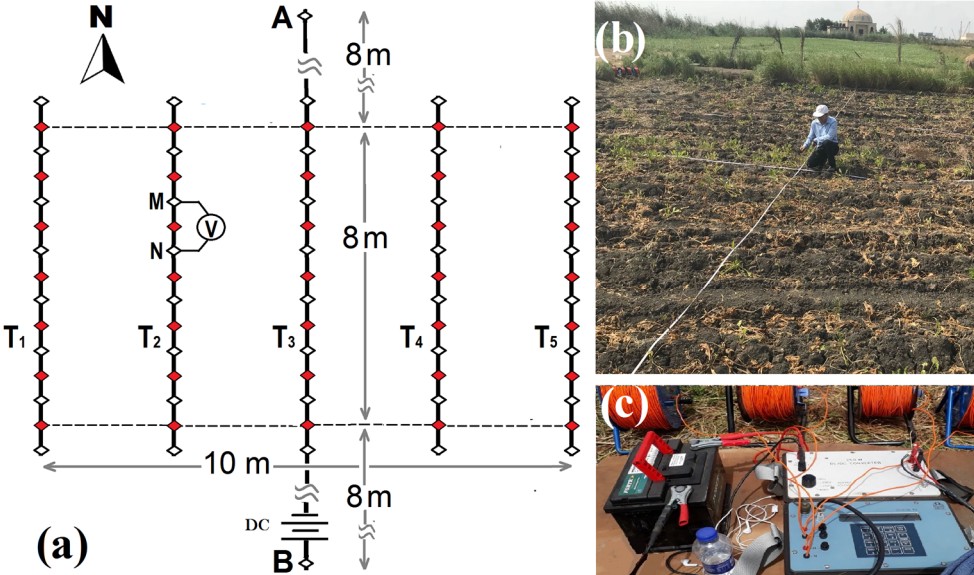

**Figure 5.** (**a**) Grid pattern used to conduct the apparent resistivity gradient (RG) survey, measurements were recorded at the midpoint, denoted by red between the two potential electrodes, (**b**) laying out the grid in the field, and (**c**) instrument used for measuring resistivity (SYSCAL-R2).

*2.3. Field Work*

2.3.1. Pilot Electrical Conductivity Survey

Prior to conducting the geoelectrical investigations according to the above, a pilot electrical conductivity (EC) survey was conducted. The WET Sensor (Delta-T Devices Ltd., Cambridge, UK) was used to measure the EC of the topsoil at the three farms. Soil samples at different depths (10–60 cm) were collected from the studied area within each farm. The first two farms (A and B) showed average EC levels ranging between 4 and 8 dS/m, while the EC of the barren farm (C) ranged between 22 and 30 dS/m. In addition to that, Table 1 shows that these samples had approximately the same composition and were characterized by their high clay content. The percentage of the soil fractions was determined using standard methods (i.e., sieve analysis and hydrometer).

**Table 1.** Summary of soil texture and density at the three investigated farms.

| Location | Depth (cm) | Sand (%) | Silt (%) | Clay (%) | Bulk Density (g·cm$^{-3}$) |
|---|---|---|---|---|---|
| | 0–10 | 3.7 | 25.8 | 70.5 | 1.8 |
| Farm A | 10–20 | 3.5 | 30.0 | 66.5 | 1.9 |
| | 20–40 | 2.1 | 32.1 | 65.8 | 1.8 |
| | 40–60 | 2.9 | 29.0 | 68.1 | 1.9 |
| | 0–10 | 4.0 | 30.8 | 65.2 | 1.8 |
| Farm B | 10–20 | 4.0 | 28.5 | 67.6 | 1.9 |
| | 20–40 | 2.4 | 31.1 | 66.5 | 1.7 |
| | 40–60 | 3.5 | 23.6 | 72.9 | 1.7 |
| | 0–10 | 3.2 | 32.7 | 64.1 | 1.9 |
| Farm C | 10–20 | 2.9 | 33.0 | 64.1 | 1.7 |
| | 20–40 | 5.6 | 25.8 | 66.6 | 1.9 |
| | 40–60 | 4.8 | 29.2 | 66.9 | 1.8 |

2.3.2. Geoelectrical Field Work

All the aforementioned geoelectrical methods were tested on Farm A. The investigation began with an ERT survey across an area characterized by poor and heterogeneous vegetation cover at the center of the farm plot. The length of the ERT profile was 10 m, and its direction was chosen to be orthogonal to the main path of the subsurface mole drains of the farm. The initial spacing between electrodes implanted along the profile was 0.4 m. To create a continuous cross-section of the subsoil, the profile was re-surveyed eight times. Each time, the spacing between electrodes increased by 0.4 m (Figure 3). This configuration sufficed to image 1.6 m depth of the soil, exceeding the depth of the mole drains. The apparent resistivity was measured at each station individually (manually), along the designed profile shown in Figure 3. This provided a good opportunity to check noise levels directly during the field acquisition and to repeat measurements. The whole set of apparent resistivity measurements was rechecked again for abnormal data points before inverting it, using both robust inversion (L1-norm) and smooth constrained least square (L2-norm) algorithms [43,44].

A set of five profiles were surveyed using the self-potential (SP) method. The survey covered a rectangular area of 10 m by 8 m. The SP profiles were spaced by 2 m, and the middle profile (P3) superimposed the exact location of the ERT profile. A reading was recorded every 40 cm along each profile. Figure 4 shows the layout of the grid used to conduct the SP survey. After correcting the SP measurements as discussed in Section 2.2, the measurements were plotted in a map showing the surface variations of the natural potential in soil.

The same rectangular area was re-surveyed using the RG method. The directions of RG surveyed profiles were chosen to be running perpendicular to the main azimuth of the detected high SP anomaly. Therefore, the length of the surveyed profiles was only 8 m long (Figure 5a). The distance between the fixed current electrodes A and B was 24 m, i.e., three times the length of the surveyed profile. Potential differences were recorded every 1 m along five profiles that lay parallel to AB and separated by 2.5 m. After calculating the $\rho_a$ for each station measured within the RG grid according to Equation (3), the results were plotted in a map that shows the lateral variation of surface apparent resistivity within the same surveyed plot.

## 3. Results and Discussion

Figure 6 compares the two ERT models produced by the different inversion algorithms. Commonly, the L1-norm method is preferred when there are high contrasts in apparent resistivity readings [62] because it is less sensitive to noisy data points compared to the L2-norm method [41]. Since the smooth constrained L2-norm method relies on minimizing the square of the differences between the measured and calculated apparent resistivity, it shows a smooth distribution of the inverted resistivity of the subsoil (Figure 6b). The robust inverse L1-norm produced a model with sharper edges than that produced by the L2-norm method (Figure 6a).

It can be noted that the L2-norm inversion gives better results than the L1-norm as it is more suitable for the subsurface geology conditions that exhibit a smooth variation [41,46]. Therefore, the L2-norm was chosen to invert the measured apparent resistivity data.

Initial examination of the SP and RG maps shown in Figure 7 reveals a significant discrepancy between the results of the methods. For instance, the elongated high anomaly in the central part of the SP map has disappeared completely from the RG map. Its location is occupied by a gradual decrease in the apparent resistivity towards the southern parts of the surveyed plot. Similarly, it was clearly observed during the fieldwork that the apparent resistivity values recorded in areas characterized by dense vegetation were always high in contrast to SP.

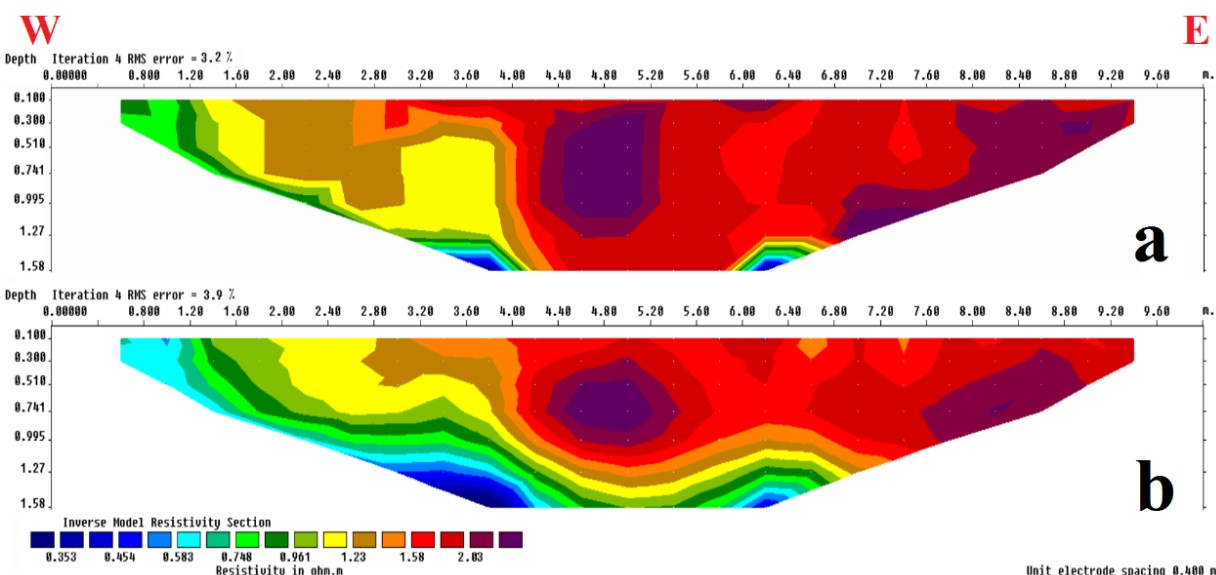

**Figure 6.** ERT models of the same section inverted using: (**a**) the robust inversion method, and (**b**) the least-squares smooth constrained method. Data were surveyed at Farm A in October 2021.

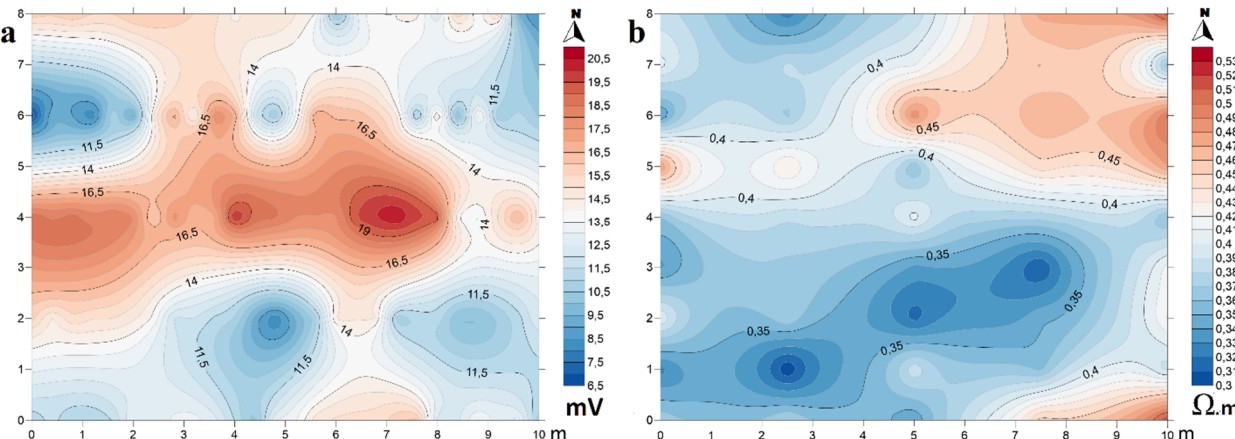

**Figure 7.** (**a**) Spontaneous potential map and (**b**) calculated apparent resistivity gradient map of the same plot surveyed at Farm A.

For better understanding, combining results of more than one geophysical method is a common practice to reduce the ambiguities in data interpretation. Thus, the results obtained from the three surveys on Farm A were compiled into one figure (Figure 8). The green and the light blue curves shown in the upper part of the figure represent lateral variations of SP and RG values, respectively. Both curves were plotted at their exact location along the ERT cross-section. This compilation revealed several important points regarding the response of each method to the changes in soil texture and structure.

First, the two curves (Figure 8) conspicuously confirm the discrepancy between the SP and RG readings, i.e., when SP shows high values at a particular location along the profile, GR correspondingly shows low values for the same location, and vice versa. Usually, the SP and GR curves diverge opposite permeable freshwater sand and converge in front of shale [63,64]. This relationship is a well-established principle used in interpreting SP and resistivity logs [65,66]. In this case, the high apparent resistivity (RG) values at 2 m, 4.8 m, and 8.2 m are accompanied by dips on the SP curve. These sites are located within regions characterized by permeable sandy soils. Conversely, the SP curve shows continuous high values along the first 1.5 m of the profile and a high peak centered at 6.4 m. At these locations, the RG readings are lower than normal, and the soil becomes more clayey.

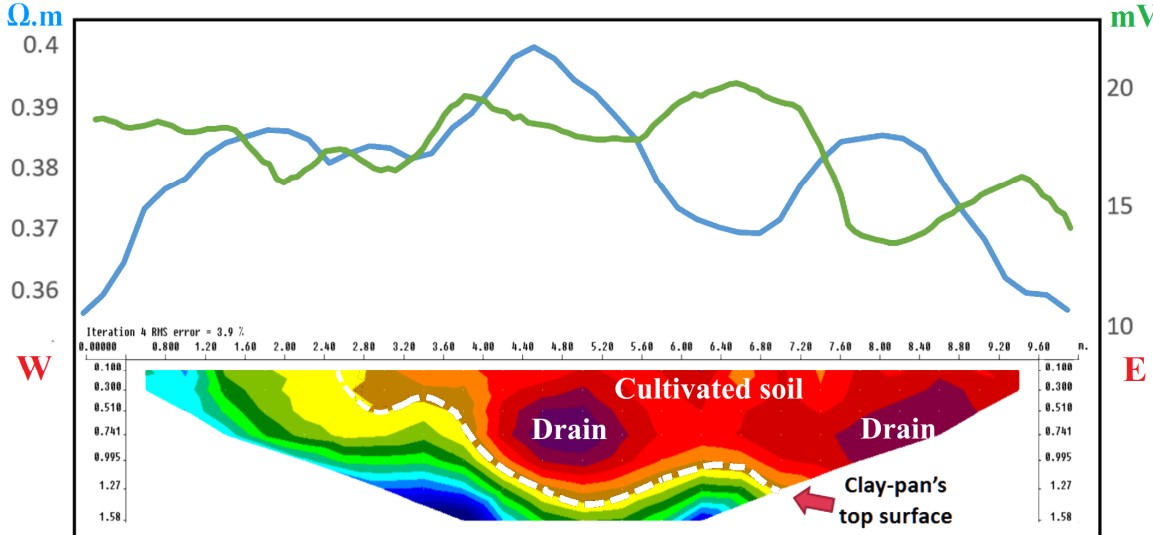

**Figure 8.** Spontaneous potential curve (green), apparent resistivity gradient curve (light blue), and the ERT model of the same profile surveyed at Farm A.

Second, there is a strong resemblance between the shape of the SP curve and the surface relief of the layer delineated by the white dashed line in the ERI model (Figure 8). As the surface of that layer approaches the ground surface of the soil, the readings on the SP curve increase. It is worth mentioning here that the abrupt changes in SP curves are commonly associated with changes in soil composition, especially with increasing clay content in the soil. Because clay particles have negative electrical charges on their surfaces, clay layers can attract and retain positively charged ions (cations) and prevent them from escaping from the soil [67]. Consequently, an electrochemical interaction occurs between clay surfaces and soil-water, especially when the latter has a different ionic activity. The electromotive force originated from this interaction causes currents to flow around the clay boundaries. Therefore, SP measurements record high positive voltages adjacent to clay layers [52,60,68], which in turn implies that this layer represents the top surface of the underlying clay-pan.

Finally, the divergence between the SP and RG curves that occurred at 4.8 m and 8.2 m is attributed to the lower clay content in the topsoil or to the change of soil texture that becomes sandier at these locations. The latter assumption fits well with the results obtained from the ERT section (Figure 8) which shows that the depth to the surface of the clay bed increases in the central and eastern parts of the farm. This allows the drainage system to function without hindrance in these parts of the farm and results in lowering the SP values due to infiltration from the drainage ditches [55]. The ERT section is also able to locate two mole drains (Figure 8). The active soil filtration nearby the locations of the drains, as the soil becomes more permeable, reduces the salinity and consequently increases its apparent resistivity. Therefore, the presence of these drains is responsible for the two high RG anomalies.

The rule of thumb in interpreting any near-surface geophysical results is that: without drilling, all interpretations are refutable. Accordingly, these results were substantiated by auguring holes along the surveyed profile. Based on the aforementioned notes, we can conclude that as depth to the clay-pan increases eastwardly, deep-rooted plants can grow in the upper permeable reworked soil. Poor vegetation on the western side of the farm can be attributed to the increase of silt and clay contents in the topsoil where the surface of the clay-pan approaches the ground surface. This has resulted in blocking the MD, thereby diminishing its efficiency in accelerating salt leaching. Moreover, soil compaction on the western side of the farm adversely affects crop production by hindering plant root growth. Compaction reduces soil aeration, lessens gaseous exchange in the soil, and limits water flow and hence drainage [69]. Thus, we can advise the farm owner to plow the western strip of the farm, whenever possible, to loosen the soil.

We plan to repeat the ERT survey along the same profile every three months. The main objective of repeating the survey is to establish the reliability of the technique in observing subsurface resistivity changes that occur on the farm after plowing. Thus, we left permanent wedges on both ends of the surveyed profile. The wedges enabled us to precisely locate the profile and repeat the ERT measurements along the same path.

It should be mentioned here that the first ERT survey was conducted in October 2021. The same profile was surveyed again in December 2021 and in April 2022. The effect of temperature changes was removed from all measurements before inversion [47,49]. The inversion of the datasets was performed using a joint inversion technique where the resistivity model of the first survey (Figure 6b) was used to constrain the inversion of the subsequent datasets. The inverted resistivity models of the time-lapse surveys are shown in Figure 9.

A qualitative comparison of the two inverted resistivity models, illustrated in Figures 6b and 9a, did not reveal any significant changes in the soil structure during the period between October and December 2021. However, it was obvious that surface plowing had slightly loosened the compacted soil on the western side of the farm, but it did not disturb the deep clay layer. As the soil became permeable for freshwater flows, its resistivity increased especially along the first 2.80 m of the profile (Figure 9a). This has improved

the functionality of the central drain, which was able to reduce the salinity of larger cross-sectional areas around its course compared to that shown in the first model (Figure 6b). This result prompted the farm owner to continue plowing the soil whenever possible. After another three months, in April 2022, the situation improved further, and the cultivated soil of the farm extended westward (Figure 9b).

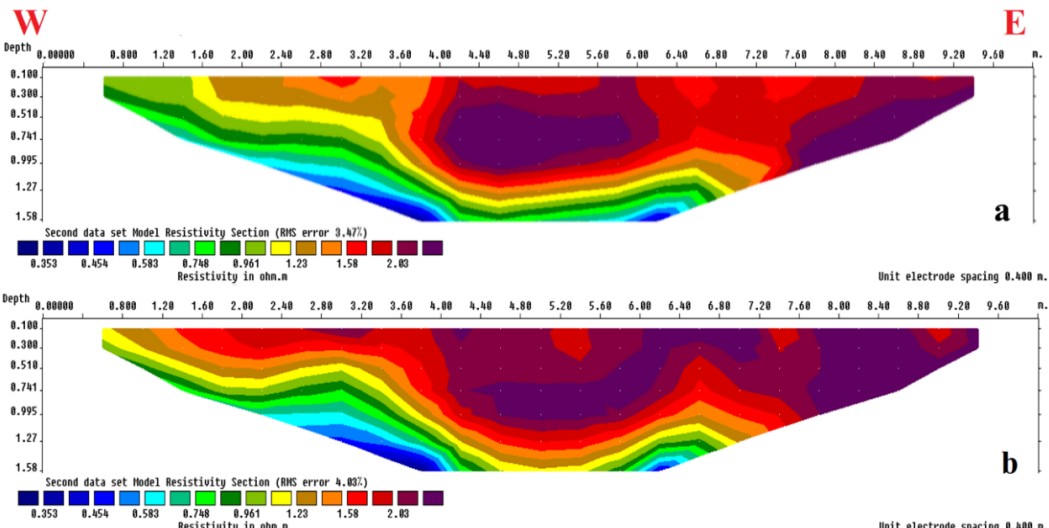

**Figure 9.** ERT models of the inverted apparent resistivity data measured along the same profile that was resurveyed in (**a**) December 2021 and (**b**) in April 2022.

The time-lapse sections shown in (Figure 10a,b) are helpful in illustrating quantitatively the changes occurring during the whole period of the survey, where the inverted resistivity model of the first survey (Figure 6b) was used as a reference to estimate the percentage changes in resistivity between the reference model and each one of the subsequent models. High percentages in these sections represent areas where maximum changes in soil resistivity occur, while low values indicate no significant changes. The figure also shows that the surface salinity of the topsoil decreased as its resistivity increased by 20% in December (Figure 10a), while it reached 80% in April (Figure 10b).

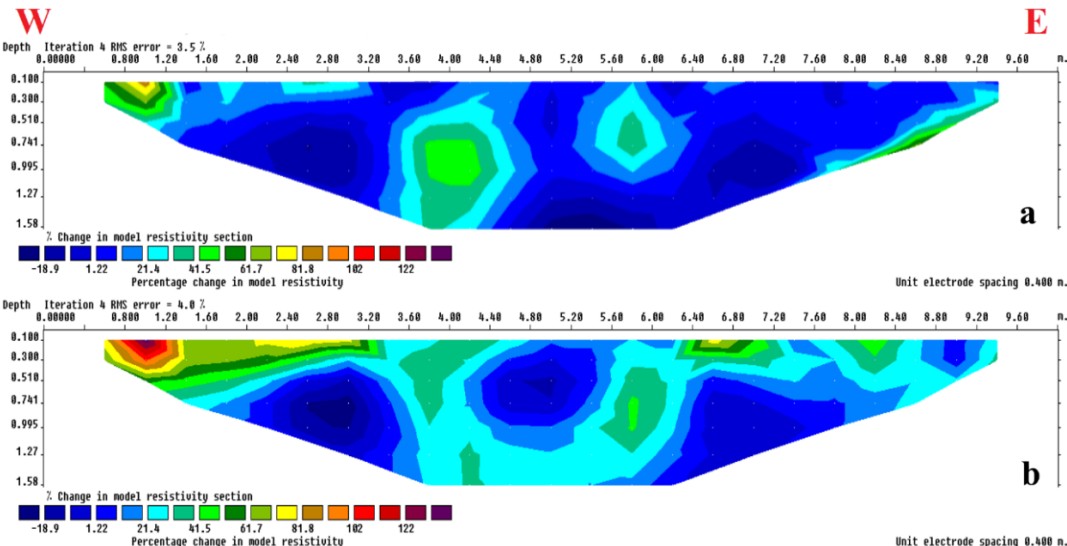

**Figure 10.** Time-lapse sections showing the percentage change in the subsurface resistivity values, at Farm A, obtained from the joint inversion between the initial model (shown in Figure 6b) as a reference with the lateral datasets collected in (**a**) December 2021 and (**b**) in April 2022.

The above experiments demonstrate the ability of the ERT method to visualize subterranean soil structure and resolve complex sources of resistivity anomalies that cannot be distinguished using RG or SP methods alone. The method was able to determine the location and depth of mole drains, trace the surface of the clay-pan layer, and estimate the thickness of the cultivable soil. This encouraged us to conduct ERT surveys directly on the other two farms (B and C).

Figure 11 shows the inverted resistivity model of the ERT survey performed at Farm B. The azimuth of the surveyed profile was chosen to run perpendicular to the direction of the buried mole drain system of the farm. High resistivity closures identify the location of the detected drains within the cultivable layer (Figure 11). Yellow arrows in the figure denote the location of these buried drains which are spaced at equal distances of 1.5 m. The average depth to the top of the clay layer is about 55 cm. This shallow depth limits the thickness of the active root zone. As a result, the thickness of the cultivable soil is not sufficient to grow vegetables with medium or deep roots such as beans, carrots, tomatoes, peas, or eggplants.

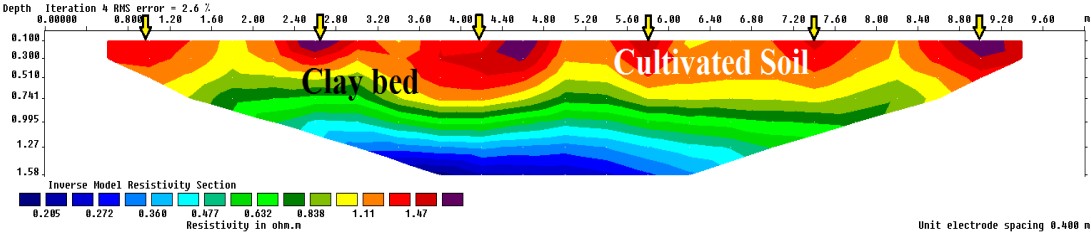

**Figure 11.** Inverted resistivity models of the ERT data conducted at plot B showing the thickness of its root zone. Yellow arrows are pointing at the locations of the drains.

As only traditional reclamation methods were practiced in Farm C, all cultivation efforts in this farm failed due to the frequent rise of water in the center of the farm (Figure 12c). EC measurements showed that the electrical conductivity of the topsoil reaches 30 dS/m. When the waterlogging stopped, we were able to conduct two ERT surveys along the same profile, parallel to the direction of the mole drains. The first survey took place in November 2021 and was then repeated in January 2022. The inverted resistivity models of the two surveys are shown in (Figure 12a,b). The first model (Figure 12a) shows the infiltration of irrigation water through the topsoil during November 2021. In January 2022, the poorly drained soil retains the infiltrated water in the deepest parts of the clay layer (Figure 12b) because the clay layer is gently inclined towards the middle part of the farm.

This inclination opposes the direction of the drainage system that was initially designed according to the surface levels of surface soil. This conflict led to the malfunctioning of the farm drainage system, which plays almost no role in reducing soil salts [70,71]. Therefore, the salinity levels of the soil and the water retained in it increased. This can be inferred from the time-lapse model (Figure 12c), as the resistivity of the saturated subsoil in Farm C has decreased by about 10%. We tried to conduct an additional ERT survey in February but were prevented by water logging (Figure 12d). Yellow arrows in Figure 12b,d indicate the location of the surface clayey soil that formed a sub-basin in the center of the farm.

It is noteworthy that designing the agricultural drainage system requires more information about the soil subsurface structure and should not be based only on the surface levels of the soil, especially for land with mild to no slope. It is recommended to image the subsurface soil structure along several arbitrary directions before designing the drainage system of any farm.

The role played by the drainage system in reducing soil salinity can be inferred through comparing the inverted ERT models of the two farms A and C. The active drainage system on the first farm (A) was able to reduce the salinity of the soil, as can be noticed from the increment of resistivity nearby the drain location (Figures 9 and 10). On the other hand,

the malfunction of the drainage system in Farm C resulted in decreasing resistivity of the soil due to increased salt content. These results confirmed the reliability of the EC measurements during the reconnaissance surveys.

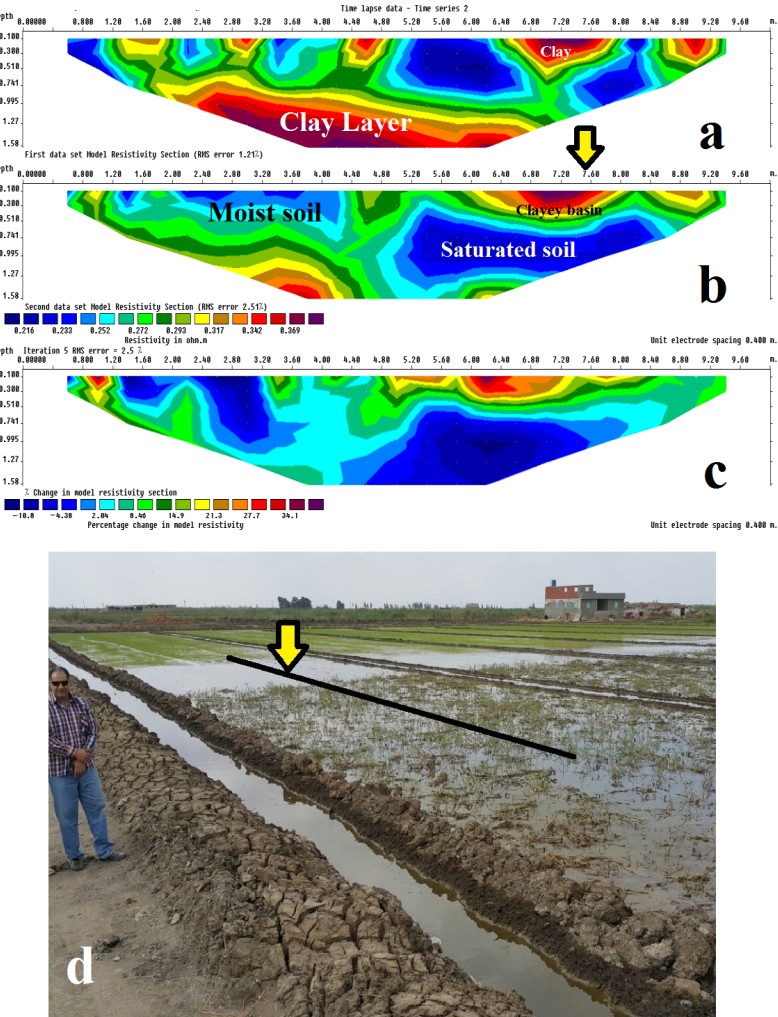

**Figure 12.** ERT models conducted at Farm C along the same profile in (**a**) November 2021 and (**b**) in January 2022. (**c**) The percentage change in the subsurface resistivity. (**d**) Waterlogs appeared on the farm in February 2022. The black line shows the location of the surveyed ERT profile. The yellow arrow points at the location of the compacted clay on the surface of the farm, which extends laterally to other plots and holds water after a rainfall.

## 4. Conclusions

Geoelectrical surveys were conducted on three newly reclaimed farms located in the northeastern corner of the Nile Delta, Egypt. The soils of all farms within this region are characterized by their high content of clay. Farmers usually rely on the traditional mole drainage (MD) on these farms. Most of the farms are currently suffering from waterlogging and salinization problems.

The present research has proven the applicability of geoelectrical methods in imaging subsurface soil structures, even for soil characterized by high clay contents and affected by salinity. Inverted ERT models can suffice to understand the subsurface structures but integrating results from SP and RG helped to understand the geology and to reduce the ambiguous interpretation that could result by using any of these methods individually. Results show that the SP measurements are mainly influenced by the contrasts in clay contents within the surface soil, while high apparent RG closures can define cultivable areas on the farm. The inverted ERT models were successful in determining the thickness

of the root zone layers, locating subsurface drains, and tracing the upper surface of the clay bed (bedrock). The shallow depth of the undisturbed clay beds reduces the productivity of the farm as it limits the thickness of the active root zone and prevents the growth of deep-rooted plants. It is necessary to consider the slopes of the subsurface clay beds before planning the drainage network of any farm, even if the farm has a flat surface. We can conclude that the efficiency of MD depends on the local subsurface soil conditions. Thus, it is also important to image the subsurface conditions of the soil periodically to monitor the dynamics of agricultural soil processes. Time-lapse ERT surveys showed their ability to monitor the accumulation of groundwater over the course of different seasons, which is the main cause of waterlogging problems on the farms.

**Author Contributions:** Conceptualization, A.A., Y.H. and T.S.; Data curation, A.A., T.A., Y.H. and T.S.; Formal analysis, A.A., T.A. and T.S.; Funding acquisition, R.B., Y.H. and T.S.; Investigation, A.A., T.A., Y.H. and T.S.; Methodology, A.A., R.B., T.A., Y.H. and T.S.; Project administration, R.B., Y.H. and T.S.; Resources, R.B., Y.H. and T.S.; Software, A.A. and T.A.; Supervision, A.A., R.B., T.A., Y.H. and T.S.; Validation, A.A., T.A. and T.S.; Visualization, A.A., R.B., T.A., Y.H. and T.S.; Writing—original draft, A.A. and T.S.; Writing—review & editing, A.A., R.B., T.A., Y.H. and T.S. All authors have read and agreed to the published version of the manuscript.

**Funding:** This research received no specific grant from any funding agency in the public, commercial, or not-for-profit sectors.

**Institutional Review Board Statement:** Not applicable.

**Informed Consent Statement:** Not applicable.

**Data Availability Statement:** The data presented in this study is available on request from the corresponding author.

**Acknowledgments:** The authors acknowledge funding for this study from the Egyptian Academy of Scientific Research and Technology and Tunisian Institution of Agricultural Research and Higher Education through the SALTFREE project (ARIMNET2, Coordination of Agricultural Research in the Mediterranean Area). The research was also supported by the Strategic Research Area: The Middle East in the Contemporary World (MECW) at the Centre for Advanced Middle Eastern Studies, Lund University, Sweden. The authors wish to thank the Center of Environmental Studies and Consultancies at Port Said University for providing the necessary equipment and permissions to conduct the field surveys.

**Conflicts of Interest:** The authors declare that there is no conflict of interest.

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
