# Peer review of "Noninvasive Monitoring of Subsurface Soil Conditions to Evaluate the Efficacy of Mole Drain in Heavy Clay Soils"

_water, doi:10.3390/w15010110_

Round 1

Reviewer 1 Report

Some information in the introduction belongs to the methodology and some literature review in the methodology should be removed to the introduction. More information on the theory of the inversion modelling needs to be added. There are several detailed comments:

Figure 5. what does the red dots mean? Please indicate. 

Line 209: "each traverse. Figure (4a) shows". Figure or Fig, please be consistent. 

Author Response

Reviewer #1 comments

Reviewer #1 comment # 1:

Some information in the introduction belongs to the methodology and some literature review in the methodology should be removed to the introduction. More information on the theory of the inversion modelling needs to be added.

Author`s response:

We are grateful to the reviewer for suggestions to improve the text. We agree with the suggestions and have now made the following changes:

  • The title ‘2. Method & material’ was moved up to line 66.
    • A new subtitle ‘1 Site description and location’ was inserted directly before the paragraph that describes the geological setting and the location of the area under investigation.

  • The following two paragraphs were moved up to the introduction section.

‘Local farmers ……….level, thereby increasing crop production.’

‘Recently, many of …………..……..inefficient drainage systems.‘

  • The following paragraph was added to provide more information on the theory of the inversion modelling

‘The inversion process aims to determine the true resistivity of the subsurface layers from the apparent resistivity pseudo-section. Thus, the process begins with constructing apparent resistivity pseudo-sections from a series of readings measured at the ground surface, by using finite-difference or finite-element methods to divide the section into many rectangular blocks with different resistivity values [40,45]. Then the Jacobian matrix of partial derivatives is calculated using Gauss-Newton or quasi-Newton methods. Finally, the process ends with solving the least least-square equation of the Gauss-Newton and quasi-Newton methods using a regularized least-square optimization method [43,44]. Two algorithms are used to constrain the regularized least-square optimization. The first is the blocky constrained least-square method (L1-norm), and the second is the smooth constrained least-square method (L2-norm) [46].’

Reviewer #1 comment # 2:

Figure 5. what does the red dots mean? Please indicate.

Author`s response:

We would like to thank the reviewer for pointing this out. We have added this text to the caption of the figure. ‘Measurements were recorded at the midpoint, denoted by red dots between the two potential electrodes’ 

Reviewer #1 comment # 3:

Line 209: "each traverse. Figure (4a) shows". Figure or Fig, please be consistent. 

Author`s response:

We agree and have now changed ‘Fig.’ to ‘Figure’ throughout the whole manuscript, according to the journal`s requirements.

Reviewer 2 Report

This article addresses an important scientific problem of agricultural soil issues. The authors used the electrical geophysical survey method to solve the problem of the inability to reform the soil and the problems of plant growth due to the difference in soil components. The results were promising and could give a good impression of the possible solutions. The research as a whole is good and may be accepted for publication in the journal after making minor corrections represented in the English language and some figures.

The following comments should be taken into consideration

·         A map must be drawn up that place the study area in relation to the worldwide map, because the study area is a local area and is unknown on the global level

·         Please add the north arrow and the coordinate systems in figure 1.b

·         The latitude and longitude coordinates of the study area should be included in the text

·         The electrode spacings for ERT should be included within the text. The authors did not mention whether they used a multielectrode cable or not and what is the name and the type of the instrument used in this study, it should be added in the material and method section

·         The “method and material” title should be changed to “methodology” because the authors did not use materials in this work.

·         In Table one, How did you separate the component of the soils to a percentage of sands, silts and clays?? Did you do sieve analysis?? Please explain.

·         Please explain the location of the ERT profiles for all figures

·         Why did the authors eliminate the description of SP and RG for the other two farms B and C

·         English language should be checked again for all the text.

·         The references should be rewritten in the same style for volumes, issues, and page numbers.

Author Response

This article addresses an important scientific problem of agricultural soil issues. The authors used the electrical geophysical survey method to solve the problem of the inability to reform the soil and the problems of plant growth due to the difference in soil components. The results were promising and could give a good impression of the possible solutions. The research as a whole is good and may be accepted for publication in the journal after making minor corrections represented in the English language and some figures. The following comments should be taken into consideration:

Reviewer #2 comment # 1:

 A map must be drawn up that place the study area in relation to the worldwide map, because the study area is a local area and is unknown on the global level.

Author`s response:

We would like to thank the reviewer thoughtful comments and efforts towards improving our manuscript. We have now added an index map in Figure 1a.

Reviewer #2 comment # 2:

Please add the north arrow and the coordinate systems in figure 1.b.

Author`s response:

We have now added the north arrow and the coordinate systems in Figure 1.b

Reviewer #2 comment # 3:

The latitude and longitude coordinates of the study area should be included in the text.

Author`s response:

We have now added Figure 1c, to show the location of the three farm plots within the newly reclaimed area. In addition, we included the latitudes and longitudes of the study area in the text. Please see Line 72 and 73 in the new revised version of the manuscript.

Reviewer #2 comment # 4:

The electrode spacings for ERT should be included within the text. The authors did not mention whether they used a multielectrode cable or not and what is the name and the type of the instrument used in this study, it should be added in the material and method section.

Author`s response:

We would like to thank the referee for spotting this.  

  • We have now included the electrode spacing for ERT. Please see lines 217 and 219 in the revised version.
  • Unfortunately, we did not have a multielectrode cable at the time of the survey. We have stated that using an implicit declaration in lines 221 – 223.
  • We also mentioned the name of the instruments used to acquire the ERT data (Lines 196 – 199, in the revised version)

Reviewer #2 comment # 5:

The “method and material” title should be changed to “methodology” because the authors did not use materials in this work.

Author`s response:

We agree with the reviewer on this comment. We have now moved the old title upward to include the site description, to follow the structure of the journal.

We inserted a new subtitle ‘2.2 Methodology’ in Line 92 of the revised version.   

Reviewer #2 comment # 6:

In Table one, How did you separate the component of the soils to a percentage of sands, silts and clays?? Did you do sieve analysis?? Please explain.

Author`s response:

We wish to express our gratitude to the reviewer for pointing this out. Yes, we did the sieve analysis and hydrometer as well. We have now included this information in the text.

Reviewer #2 comment # 7:

Why did the authors eliminate the description of SP and RG for the other two farms B and C.

Author`s response:

As Farm A showed a higher variability in salinity levels and crop production as compared to other plots, the SP and RG measurements were conducted. Also, in order to save space, the results of Farm A were only introduced.

Reviewer #2 comment # 8:

English language should be checked again for all the text.

Author`s response:

We would like to thank the reviewer for pointing this out to improving our manuscript. A native English speaker have now checked all text.

Reviewer #2 comment # 9:

The references should be rewritten in the same style for volumes, issues, and page numbers.

Author`s response:

All references were modified according to the format of Water journal using EndNote.
